# Clinical Testing for Mismatch Repair in Neoplasms Using Multiple Laboratory Methods

**DOI:** 10.3390/cancers14194550

**Published:** 2022-09-20

**Authors:** Richard K. Yang, Hui Chen, Sinchita Roy-Chowdhuri, Asif Rashid, Hector Alvarez, Mark Routbort, Keyur P. Patel, Raja Luthra, L. Jeffrey Medeiros, Gokce A. Toruner

**Affiliations:** 1Department Pathology, The University of Texas MD Anderson Cancer Center, Houston, TX 77030, USA; 2Department of Hematopathology, The University of Texas MD Anderson Cancer Center, Houston, TX 77030, USA

**Keywords:** mismatched repair deficiency, microsatellite instability, solid tumors, next-generation sequencing, immunohistochemistry

## Abstract

**Simple Summary:**

There are limited studies that incorporate genetic/epigenetic alterations into the assessment of the microsatellite instability (MSI) and mismatch repair (MMR) determination of tumors. While MSI and MMR testing are part of the screening for the eligibility to employ immune checkpoint inhibitor (ICI) therapy, data from next-generation sequencing (NGS) are not used in the current practice. For most neoplasms, IHC- and PCR-based MSI testing results are concordant. However, for neoplasms with major discordance in IHC and MSI testing, the addition and integration of next-generation sequencing (NGS) results and *MLH1* promoter methylation analyses can be beneficial for resolving borderline cases, thereby facilitating patient management.

**Abstract:**

**Background:** A deficiency in DNA mismatch repair function in neoplasms can be assessed by an immunohistochemical (IHC) analysis of the deficiency/loss of the mismatch repair proteins (dMMR) or by PCR-based methods to assess high microsatellite instability (MSI-H). In some cases, however, there is a discrepancy between the IHC and MSI analyses. Several studies have addressed the issue of discrepancy between IHC and MSI deficiency assessment, but there are limited studies that also incorporate genetic/epigenetic alterations. **Methods:** In this single-institution retrospective chart-review study, we reviewed 706 neoplasms assessed between 2015 and 2021. All eligible neoplasms were assessed by IHC testing, MSI analysis by PCR-based assay, and tumor-normal paired next-generation sequencing (NGS) analysis. Eighty percent of neoplasms with MLH1 protein loss had a concurrent *MLH1* promoter methylation analysis. Mutation data for MMR genes, IHC, MSI analysis, and tumor histology were correlated with each other. **Results:** Fifty-eight (8.2%) of 706 neoplasms had MSI-H by PCR and/or dMMR by IHC. Of the 706 analyzed neoplasms, 688 neoplasms (98%) had concordant results: MSI-H/dMMR (n = 44), microsatellite-stable (MSS)/proficient MMR (pMMR) (n = 625), and MSI-Low (L)/pMMR (n = 19). Of the remaining 18 neoplasms, 9 had a major discordance: MSS/loss of MSH2 and MSH6 (n = 3), MSS/loss of MSH6 (n = 2), MSS/Loss of MLH1 and PMS2 (n = 1), and MSI-High/pMMR (n = 3). In total, 57% of cases with dMMR and 61% of cases with MSI-H had a null mutation of an MMR gene mutation (or methylation of the *MLH1* promoter), whereas this figure was 1% for neoplasms with a normal IHC or MSI pattern (*p* < 0.001). Among 9 cases with major discordance between MSI and IHC, only 3 cases (33%) had an underlying genetic/epigenetic etiology, whereas 37 (76%) of 49 cases with MSI-H and/or dMMR and without major discordance had an underlying genetic abnormality (*p* = 0.02). **Discussion**: For most neoplasms, IHC and PCR-based MSI testing results are concordant. In addition, an underlying genetic abnormality (a null mutation of an MMR gene or *MLH1* promoter methylation) was attributable to dMMR and/or MSI-H findings. For neoplasms with major discordance in IHC and MSI testing, the addition and integration of NGS results and *MLH1* promoter methylation analyses can be beneficial for resolving borderline cases, thereby facilitating patient management.

## 1. Introduction

The mismatch repair (MMR) system is highly important for the protection of genomic integrity. The MMR system specifically repairs small mismatches and small insertion/deletions (indels). It is a highly evolutionary conserved system. When the MMR system is deficient, deleterious DNA damage accumulates, which results in an increased number of mutations in the cell and can result in neoplasia [1].

Traditionally, MMR system deficiency was assessed to screen patients for Lynch syndrome (LS), a hereditary cancer predisposition syndrome characterized by the germline mutation of four cancer repair genes, *MLH1*, *PMS2*, *MSH2*, and *MSH6*. Patients with LS are most prominently afflicted with colorectal and endometrial neoplasms, but ovarian, gastric, pancreatobiliary, urothelial, small bowel, and CNS tumors also can be observed in these patients [2]. It should be noted that MMR deficiency is not necessarily indicative of LS, and sporadic cases with MMR deficiency are not uncommon. MMR-deficient colorectal cancers (CRCs) have a better prognosis than MSS CRCs, and the presence of mismatch repair deficiency guides adjuvant therapy selection in dMMR and MSI-H CRC patients [3,4]. The other clinical reason for assessing these biomarkers is as part of the screening for the eligibility to employ immune checkpoint inhibitor (ICI) therapy. Since patients afflicted with neoplasms harboring mismatch repair deficiency demonstrate increased responsiveness to ICI [5,6,7], mismatched repair deficiency is regarded as a biomarker for using ICI, and the universal screening of all tumors for mismatch repair deficiency is recommended. 

MMR deficiency is primarily assessed using two different methods. One method is immunohistochemistry (IHC), and the other is microsatellite instability (MSI) testing by polymerase chain reaction (PCR). With IHC testing, the absence of the expression of MLH1, PMS2, MSH2, and MSH6, four key proteins of the MMR system, is assessed. If all four proteins are expressed, the MMR system is reported to be proficient (or intact). If the loss of the expression of at least one of the MMR proteins is observed, this finding most often indicates the presence of mismatch repair deficiency (dMMR) by IHC [8] and therefore the loss of mismatch repair function.

Microsatellite instability (MSI) is regarded as the phenotypic expression of mismatch repair defects in the genome. Microsatellites are repetitive sequences of DNA with a repeating unit size of less than six bases. Microsatellites are scattered throughout the coding and noncoding parts of the genome. Because of their repetitive nature, these sequences are susceptible to shortening or lengthening and subsequent misalignment due to DNA polymerase slippage and are prone to mutation during DNA replication. The mismatch repair system prevents the accumulation of these mismatch errors. When the system is deficient, mismatch errors accumulate, primarily in microsatellite regions, and result in microsatellite instability [9,10]. 

As discussed above, the downstream effect of mismatch repair deficiency is microsatellite instability as well as subsequent genetic mutations that can drive neoplasia, whereas the upstream cause of MSI-H is mutations in *MLH1*, *MSH2*, *MSH6,* and *PMS2* and/or *MLH1* promoter methylation. In addition, 3’*EPCAM* deletions can lead to mismatch repair deficiency secondary to the hypermethylation of *MSH2* [11]

The accurate determination of mismatch repair deficiency is obviously of critical importance. Several studies have addressed the issue of concordance [12,13,14,15,16,17,18,19,20,21,22,23,24,25,26,27,28,29,30,31,32,33,34] between MSI and IHC for assessing mismatch repair deficiency. However, there are a limited number of studies [13,28,29,30,31,34] that also take into account somatic gene mutations. We had two major aims in this study. First, we assessed the concordance/discordance between MSI and IHC testing and asked the following questions: “1. What is the concordance rate between MSI and IHC?”, “2. Does this concordance rate change according to neoplasm type?”, and “3. Are certain IHC loss patterns more likely to be discordant with MSI?”. Secondly, we assessed the relationship between gene mutations, IHC, and MSI and asked the following questions: “4. What is the rate of positive IHC and MSI results attributable to genetic/epigenetics alterations?” and “5. Are some types of gene mutations (and epigenetic alterations) more likely result in loss of mismatched proteins and MSI?”, and “6. Can we use gene mutation testing results to resolve discordance?”.

To address these questions, we performed this single-center retrospective study on 706 neoplasms with concurrent IHC testing, MSI analysis, and tumor-normal paired next-generation sequencing (NGS) analysis.

## 2. Materials and Methods

### 2.1. Case Selection

Clinical reports from 706 neoplasms assessed between 2015 and 2021 (345 gastrointestinal, 142 genitourinary, 192 gynecologic, and 27 miscellaneous) were reviewed from the MD Anderson Cancer Center (MDACC) electronic medical records. All neoplasms were assessed by IHC testing, PCR-based MSI analysis, and tumor-normal paired next-generation sequencing (NGS) analysis. Clinical data and pathologic findings were reviewed, and patient-level data are available in Appendix A. Data were collected following the institutional review board guidelines, which were in accordance with the Declaration of Helsinki.

### 2.2. Microsatellite Instability (MSI) Analysis

Genomic DNA was extracted from microdissected paraffin-embedded tumor sections using the Arcturus PicoPure DNA Extraction Kit (ThermoFisher Scientific, Waltham, MA, USA) as well as the Agencourt AMPure XP PCR purification system (Beckman Coulter Life Sciences, Brea, CA, USA), and peripheral blood DNA was extracted using the Maxwell RSC Blood DNA Kit (Promega, Madison, WI, USA) according to the manufacture’s guidelines and as part of a routine clinical workflow. Genomic tumor and normal control DNA were analyzed by routine clinical PCR-based methodology. Briefly, Go Taq polymerase (Promega), 5X Colorless GoTaq Flexi Buffer (Promega), 10 mM dNTP mix, 25 mM MgCl_2_, and fluorescent dye (ROX, 6-FAM, NED, or VIC)-labeled primers were used. The primer sequences can be found in Appendix A. The PCR reactions with cycle conditions were as follows: activation of enzyme (95 °C, 7 min); 3 cycles (94 °C, 1 min; 58 °C, 30 s; 72 °C, 45 s); 42 cycles (93 °C, 45 s; 54 °C, 30 s; 72 °C, 40 s); 72 °C, 45 min; 4 °C hold. PCR was followed by a 1:10 to 1:80 dilution of PCR products, the addition of HIDI-formamide, denaturation (95 °C, 5 min), and capillary electrophoretic detection using the ThermoFisher Applied Biosystems 3730XL DNA Analyzer and Gene-Mapper software. A panel of seven microsatellite markers (BAT25, BAT26, BAT40, D2S123, D5S346, D17S250, and TGFBR2) was evaluated to detect changes in the number of microsatellite repeats in tumors compared with normal tissue. BAT25, BAT26, D2S123, D5S346, and D17S250 are recommended by the National Cancer Institute (NCI) Workshop on Microsatellite Instability for Cancer Detection and Familial Predisposition. Tumors were classified as MSI-High: at least three of the seven microsatellite markers showed instability; MSI-Low: one or two of the seven markers showed instability, and microsatellite-stable (MSS): none of the seven markers showed instability. 

### 2.3. IHC Analysis

An immunohistochemical analysis was performed in accordance with a routine clinical protocol. The Ventana MMR RxDx panel with the following primary antibody clones and dilutions was used: MLH1 (G168-728, 1:300, Cell Marque, Rocklin, CA, USA), PMS2 (A16-4, 1:125 BD Biosciences, San Jose, CA, USA), MSH2 (FE11, 1:100, Calbiochem, San Diego, CA, USA), and MSH6 (44, 1:300, BD Biosciences, San Jose, CA, USA). IHC findings were classified as intact: pMMR, all four MMR proteins have normal strong nuclear expression in the tumor and surrounding cells; dMMR: the loss of nuclear expression of at least one MMR protein in tumor cells only; and equivocal: questionable results from IHC assessment.

### 2.4. MLH1 Methylation

DNA was extracted from microdissected formalin-fixed paraffin-embedded (FFPE) tissue and treated with bisulfite to convert unmethylated cytosine to uracil using the Zymo Research EZ DNA Methylation-Gold Kit (Irvine, CA, USA) and the following conditions: 98 °C, 10 min; 64 °C, 2.5 h; 4 °C, hold. Methylation-specific PCR amplification, targeting the *MLH1* promoter, was performed with fluorescently labeled primer sequences. Briefly, Platinum Taq DNA Polymerase with 10×PCR Buffer and 50 mM MgCl_2_ (Invitrogen), 10 mM dNTP (ABI), and fluorescent dye (6-FAM)-labeled primers were used. Primer sequences were as follows: 5′-GATAGCGATTTTTAACGC-3′ (forward methylated), 5′-AGAGTGGATAGTGATTTTTAATGT-3′ (forward unmethylated), and 5′-6FAMTCTATAAATTACTAAATCTCTTC-3′ (reverse). The PCR reaction cycle conditions were as follows: activation of enzyme (95 °C, 2 min); 45 cycles (95 °C, 30 s; 55 °C, 30 s; 72 °C, 1 min); 72 °C, 10 min; 4 °C hold. The PCR products were visualized by capillary electrophoresis (3730XL DNA Analyzer and Gene-Mapper software) to detect amplicons corresponding to methylated (87 bp) and unmethylated (92 bp) *MLH1* promoter regions.

### 2.5. Definition of Concordance, Minor Discordance, and Major Discordance

Concordance: When both the MSI and IHC results are in agreement for mismatch repair deficiency: MSS/intact, MSI-H/dMMR, or MSI-L/equivocal.

Major Discordance: When the MSI and IHC results are in disagreement for mismatch repair deficiency: MSS/dMMR or MSI-H/intact.

Minor Discordance: When either the MSI or IHC results in an intermediate “grey zone”: MSS-L/dMMR; MSI-L/intact, MSS/equivocal, or MSI-H/equivocal. 

### 2.6. Classification of Genetic Testing Results in MLH1, MSH2, MSH6, and PMS2

Null pathogenic variant: pathogenic nonsense, frameshift, canonical ±1 or 2 base pair splice site, initiation codon, single- or multi-exon deletion variants in MMR genes, which fulfill pathogenic variant strength 1 (PVS1) definition by ACMG/AMP criteria [35]. Pathogenic variant: a “non-null” variant that is designated as pathogenic in ClinVar (https://www.ncbi.nlm.nih.gov/clinvar/ (accessed on 11 July 2022). Variant of uncertain clinical significance (VUS): a variant that is designated as VUS or conflicting data or without any record in ClinVar.

### 2.7. Statistical Methods

The Minitab 13.0 software package was used for Chi-square and Fisher exact tests, when appropriate.

## 3. Results

### 3.1. Mismatch Repair Deficiency with PCR and IHC in Solid Neoplasms

Among 706 neoplasms, colorectal carcinoma, endometrial carcinoma, and urothelial carcinoma constituted 552 (78%) of the neoplasms in this study. In total, 58 (8.2%) of 706 neoplasms were MSI-High by PCR and/or dMMR by IHC, while 53 (7.5%) neoplasms showed dMMR by IHC and 49 (6.9%) neoplasms showed MSI-High by PCR (Table 1).

Typical MMR loss patterns, such as the loss of MLH1/PMS2 and the loss of MSH2/MSH6 made up of 42 (79%) of the 53 cases with dMMR loss, but atypical dMMR loss patterns, such as the loss of MLH1/MSH2 and the isolated losses of MSH2, MSH6, and PMS2, were also observed (Table 2). Overall, endometrial and prostate carcinomas had the highest rates (15–18%) of mismatch repair deficiency (see Table 1 for a summary and Appendix A for case-level data).

### 3.2. IHC and MSI Analyses Are Concordant in Most Neoplasms

In total, 688 (97.4%) of 706 neoplasms had concordant results (i.e., MSI-H/dMMR, MSS/pMMR, and MSI-L/pMMR). Of the remaining 18 neoplasms, 9 (1.3%) had a major discordance and 9 (1.3%) had a minor discordance between IHC and MSI (Table 3). (Case-level data containing IGV traces, capillary electrophoresis gel images, and IHC images for two cases with major discordance and two cases with minor discordance are presented in Appendix A).

In total, 44 of 49 (90%) neoplasms with MSI-High were concordant with dMMR. Five (10.2%) cases, however, had discordant results. Three (6.1%) cases with MSI-High/pMMR had major discordance, and two (4.1%) cases with MSI-High/indeterminate had minor discordance (Table 3).

In total, 44 (83%) of 53 neoplasms with dMMR had concordant MSI-High results, whereas 9 (17%) neoplasms had discordant results. Six (11.3%) cases had major discordance, and three cases (5.6%) had minor discordance. For two cases with minor discordance, MLH1 promoter methylation was also observed (Table 3). 

In total, 26 (90%) of the 29 neoplasms with the loss of MLH1/PMS2 and 10 of the 13 neoplasms with loss of MSH6/MSH2 had concordant MSI-High results. All three (100%) of the three neoplasms with the isolated loss of MSH2 and all three (100%) of the three neoplasms with the isolated loss of PMS2 had concurrent MSI-High results. For the isolated losses of MSH2 and PMS2, this rate was 100%. However, only one (25%) of the four neoplasms with isolated MSH6 loss had concurrent MSI-High results (Table 2).

When stratified by tumor histology, the discordance rate shows some mild variation, with colorectal tumors and urothelial tumors being the lowest and neuroendocrine and prostate neoplasms being the highest (Table 1).

### 3.3. Null Mutations of MMR Genes and Methylation of the MLH1 Gene Promoter Are Attributable to Most Incidents of dMMR

In total, 30 (57%) of 53 neoplasms with dMMR had a somatic null mutation of an MMR gene or methylation of the promoter of the *MLH1* gene, whereas only 7 (1.1%) of 653 with normal/indeterminate dMMR IHC patterns had a somatic null MMR gene mutations or *MLH1* methylation (*p* < 0.001) (Table 4) (Case-level data can be seen in the Appendix A).

In total, 20 (69%) of 29 neoplasms with the loss of MLH1/PMS2 expression had an attributable epigenetic *MLH1* methylation and/or a somatic null *MLH1* mutation (Table 4). Two cases had a somatic null mutation (*MLH1* or *MSH6*) in addition to *MLH1* methylation (Appendix A). Among the remaining cases, one patient had a germline pathogenic *MSH6* null gene mutation (Table 4). 

In total, 6 (46%) of the 13 neoplasms that had a loss of MSH2/MSH6 expression had an attributable somatic pathogenic *MSH2* null or *MSH6* null mutation (Table 4). One of the five cases with a somatic *MSH2* null mutation also had an *MSH2* germline pathogenic null variant (Appendix A).

For 11 neoplasms with atypical IHC patterns (isolated loss of MMR proteins or combined MLH1/MSH2 loss), all three (100%) of the three cases with isolated PMS2 loss had an underlying *PMS2* mutation. Two (66%) of the three cases with an isolated loss of MSH2 expression had an underlying germline pathogenic null *MSH2* variant. Interestingly, the other case with isolated MSH2 loss had a somatic *MSH6* VUS. Only one of the four (25%) cases with an isolated loss of MSH6 expression had an attributable somatic null *MSH6* mutation. One case with a unique loss of MLH1 and MSH2 expression had a somatic VUS on *MSH2* (Table 4).

Among 653 patients without overt dMMR loss, 598 (92%) patients had no somatic or germline null mutations, variants of potential clinical significance, or *MLH1* promoter methylation (Appendix A). Among the remaining 55 patients with germline or somatic mutations or MLH1 promoter methylation, 46 (84%) patients had only germline or somatic VUS mutations. The remaining cases had at least one somatic pathogenic/likely pathogenic mutation or *MLH1* methylation: *MLH1* methylation *MSH2* null, *MSH6* null, MSH2 LP, and PMS2 null (n = 1) (Table 4, Appendix A). 

### 3.4. Most Incidents of MSI-High Can Be Attributed to Null Mutations of MMR Genes and MLH1 Gene Promoter Methylation

In total, 30 (61%) of 49 neoplasms with MSI-High status by PCR had a somatic null mutation of an MMR gene or methylation of the *MLH1* gene promoter, whereas only 5 (0.78%) of 634 neoplasms with MSS patterns had a somatic null MMR gene mutation or *MLH1* methylation (*p* < 0.001) (Table 4).

Among the 49 cases with MSI-High, 18 (37%) cases had *MLH1* promoter methylation (Table 4). In addition to *MLH1* promoter methylation, 3 of these 18 neoplasms had either a somatic pathogenic *MSH6* null, *MLH1* null, or *PMS2* VUS mutation (Appendix A).

Somatic null mutations of MMR genes were observed in 14 (29%) of 49 cases with MSI (Table 4). Two cases had *MLH1* null mutations. Six cases had *MSH2* null mutations, and two of these six cases also had a germline MSH2 pathogenic null variant. Four cases had a somatic pathogenic *MSH6* null mutation, and two also had a germline *MSH6* pathogenic null variant. Two cases had a somatic pathogenic *PMS2* null mutation; one of these two cases had an additional germline pathogenic *PMS2* null variant (Appendix A).

In total, 6 (12%) of 49 MSI-High cases had somatic missense MMR gene mutations only. One case had a likely pathogenic somatic *MSH2* mutation. The remining five cases had somatic VUS mutations in the *MSH2*, *MSH6,* and *PMS2* genes (Table 4).

Among 23 cases with an MSI-Low phenotype by PCR, 2 cases had *MLH1* methylation, 2 cases had somatic *MSH2* VUS mutations, and 1 case had a germline *MLH1* VUS mutation (Table 4).

Among 634 patients with MSS, 585 (92%) patients had no somatic or germline null mutations or variants of potential clinical significance. Among the remaining 49 patients with mutations, 29 (59%) had only a germline VUS mutation, and 5 (10%) of 49 mutated patients with MSS had somatic pathogenic *MSH2* null, *MSH6* null, and PMS2 null mutations (Table 4, Appendix A).

### 3.5. Most of the Neoplasms with Major Discordance Do Not Have Documented Underlying Genetic/Epigenetic Pathogenic Abnormality

Among 9 cases with major discordance between MSI and IHC, only 3 cases (33%) had an underlying pathogenic genetic/epigenetic MMR gene abnormality (Table 3), whereas 37 (76%) of 49 cases with MSI-H and/or dMMR and without major discordance had an underlying genetic abnormality (*p* = 0.02) (Appendix A). 

## 4. Discussion

The classical paradigm for clinical mismatch repair deficiency asserts that dMMR is attributable to the loss of MLH1/PMS2 in most cases, to the loss of MSH2/MSH6 in a minority of cases, and to the isolated loss of individual MMR proteins in very rare cases. According to this paradigm, detected MMR protein loss should be nearly perfectly associated with microsatellite instability (MSI-High). 

In our cohort, the results are in accordance with the accepted conventional wisdom, as the IHC and MSI results were concordant in most cases. Only 1% of the cases had a major discrepancy (Research Question 1). The discordance rate shows some mild variation, with colorectal tumors and urothelial tumors being the lowest and neuroendocrine and prostate neoplasms being the highest. However, this variation was not significant (Research Question 2), and it did not deviate from the reported rates of concordance in the literature [1]. 

Unique IHC staining patterns, such as the loss of MLH1/MSH2 and isolated losses MSH2 and PMS2, did not have a major discordance with the MSI analysis (they were MSI-High) (with a notable exception of one case with pMMR and MLH1 methylation). However, 3 (23%) of 13 cases with losses of MSH2/MSH6 and 3 (75%) of 4 cases with isolated losses of MSH6 had MSS or were MSI-Low. It is remarkable that most of the discordant cases involved MSH6 protein (Research Question 3). There may be both biological and technical reasons behind this discrepancy with MSH6 staining. The biological reasons might include the mitigation of the absence of MSH6 by MSH3, as there is a functional redundancy between these two molecules. Potentially, MSH2–MSH3 dimers may replace the function of the MSH2–MSH6 dimers [36]. Technical reasons might include the known weak staining of tumor nuclei by MSH6 and the subclonal nature of the IHC MSH6 findings [37,38].

About 60% of the tumors with dMMR/MSI-High status had documented underlying genetic/epigenetic alterations of MMR genes (Research Question 4). More than 90% of these alterations were either null mutations leading to the premature termination of the transcription of MMR genes or *MLH1* promoter methylation that suppressed transcription. In addition, compared to cases with the concordant dMMR/MSI-High phenotype, the cases with a major discordance had a significantly lower rate of underlying genetic/epigenetic abnormality. Our data unequivocally demonstrate that null mutations leading to the premature termination of the transcription of MMR genes or *MLH1* promoter methylation, which suppress transcription, are the major root causes of dMMR/MSI-High status (Research Question 5).

To resolve discrepancy between IHC and MSI results, when present, the detection of pathogenic null mutations of MMR genes or MLH1 methylation is very helpful, as these alterations establish the genetic basis of the observed positive finding (dMMR or MSI-High) and provide reassurance that the observed positive finding is a true positive. In our cohort, one third of the cases with major or minor discordance had an underlying null mutation or MLH1 methylation (Research Question 6). The clinical utility for resolving discrepancies is much more limited for point mutations with uncertain clinical significance (VUS).

In our cohort, we observed several cases with germline and somatic variations of uncertain clinical significance (VUS) in these mismatch repair genes. Deciding how to interpret these variants is an important clinical challenge. One argument might be for VUS mutations, negative MSI, and IHC testing (i.e., concordant negative findings), which should be the end of an interpretation odyssey, and that these variants should be downgraded to a likely benign category. A counter argument might be that these variants are likely associated with increased missense mutations, which may not be fully assessed by MSI or IHC testing and thus could lead to increased tumor mutation burden. Therefore, it may be premature to write-off missense VUS germline and somatic variants in these genes. 

The main limitation of this work is that it is a single-center retrospective study based on signed clinical testing results for MSI, IHC, and tumor-normal paired NGS genetic testing for mismatch repair genes. The distribution of the cases may reflect a unique practice caseload in our institution rather than a general trend for all MSI/dMMR testing. Furthermore, this study is only based on clinically reported testing results (primary data were not reviewed). Finally, one may argue that, in the days of large NGS panels with built-in MSI and TMB results from large reference or academic laboratories, capillary-electrophoresis-based MSI analysis is somewhat outdated. However, there are still many laboratories performing testing with capillary-electrophoresis-based MSI analysis and small NGS panels. For laboratories running large NGS panels, when MSI and TMB data are available it may be a good practice to correlate the mutations of the MMR genes with these parameters. 

This study is one of the largest case series reported with the aim of correlating MSI, IHC, and NGS genetic testing. Our results demonstrate the importance of integrating the NGS genetic testing results of mismatch repair genes to help explain atypical IHC patterns and resolve the discordance between MSI and IHC during regular clinical practice.

## 5. Conclusions

For most neoplasms, IHC and PCR-based MSI testing results are concordant. In addition, an underlying genetic abnormality (a null mutation of an MMR gene or *MLH1* promoter methylation) was attributable to dMMR and/or MSI-H findings. For neoplasms with major discordance of IHC and MSI testing, the addition and integration of NGS results and *MLH1* promoter methylation analyses can be beneficial for resolving borderline cases, thereby facilitating patient management.

## Figures and Tables

**Table 1 cancers-14-04550-t001:** Distribution of cases according to MSI and IHC statuses and tumor histology of concordant and discrepant IHC/MSI results based on tumor histology.

	Microsatellite Instability (MSI)			Immunohistochemistry (IHC)			Cases with Discrepancy?		Cases with Major Discrepancy?		Cases with Minor Discrepancy?	
Cases (%)												
	MSI-High	MSI-Low	MSS	dMMR	Indeter.	Proficient	Yes	No	Yes	No	Yes	No
**Gastrointestinal (n = 340)**	**16(5)**	**13(4)**	**31(91)**	**18(5)**	**1(0)**	**321(95)**	**3(4)**	**337(96)**	**1(0)**	**339(100)**	**2(0)**	**338(100)**
Colorectal Carcinoma (n = 316)	15(5)	10(3)	291(92)	16(5)	1(0)	299(95)	2(1)	314(99)	0(0)	316(100)	2(1)	314(99)
Pancreatic Carcinoma (n = 14)	1(7)	1(7)	12(86)	2(14)	0(0)	12(86)	1(7)	13(93)	1(7)	13(93)	0(0%)	14(100)
Miscellaneous (n = 10)	0(0)	2(20)	8(80)	0(0)	0(0)	10(100)	0(0)	10(100)	0(0)	10(100)	0(0%)	10(100)
**Gynecologic (n = 189)**	**21(11)**	**7(4)**	**161(85)**	**21(11)**	**3(2)**	**165(87)**	**8(4)**	**181(96)**	**3(2)**	**186(98)**	**5(2)**	**184(98)**
Endometrial Carcinoma (n = 133)	20(15)	5(4)	108(81)	20(15)	1(1)	112(84)	6(7)	127(93)	3(2)	130(98)	3(2)	130(98)
Ovarian Carcinoma (n = 42)	0(0)	2(5)	40(95)	0(0)	2(5)	40(95)	2(5)	40(95)	0(0)	42(100)	2(5)	40(95)
Miscellaneous (n = 14)	1(7)	0(0)	13(93)	1(7)	0(0)	13(93)	0(0)	14(100)	0(0)	14(100)	0(0)	14(100)
**Genitourinary (n = 144)**	**11(8)**	**3(2)**	**130(90)**	**10(7)**	**1(1)**	**133(92)**	**3(2)**	**141(98)**	**2(1)**	**142(9)**	**1(1)**	**143(99)**
Urothelial Carcinoma (n = 103)	6(6)	1(1)	96(93)	5(5)	0(0)	98(95)	1(2)	102(99)	1(1)	102(99)	0(1)	103(99)
Prostate Adenocarcinoma (n = 27)	5(18)	1(4)	21(78)	5(18)	1(4)	21(78)	2(11)	25(93)	1(4)	26(96)	1(4)	26(96)
Miscellaneous (n = 14)	0(0)	1(7)	13(93)	0(0)	0(0)	14(100)	0(7)	14(100)	0(0)	14(100)	0(0)	14(100)
**Endocrine (n = 9)**	**0(0)**	**0(0)**	**9(100)**	**1(11)**	**0(0)**	**8(89)**	**1(11)**	**8 (89)**	**1(11)**	**8(89)**	**0(0)**	**9(89)**
Adrenocortical carcinoma (n = 4)	0(0)	0(0)	4(100)	0(0)	0(0)	4(100)	0(0)	4(100)	0(0)	4(100)	0(0)	4(100)
Neuroendocrine tumors (n = 5)	0(0)	0(0)	5(100)	1(20)	0(0)	4(80)	1(20)	4(80)	1(20)	4(80)	0(20)	5(100)
**Unknown primary (n = 14)**	**1(7)**	**0(0)**	**13(93)**	**3(21)**	**0(0)**	**11(79)**	**2(14)**	**12(86)**	**2(14)**	**12(86)**	**0(14)**	**14(100)**
(Adenocarcinoma)												
**Other neoplasm (n = 10)**	**0(0)**	**0(0)**	**10(100)**	**0(0)**	**1(10)**	**9(90)**	**1(10)**	**9(10)**	**0(0)**	**10(100)**	**1(10)**	**9(10)**
**All (n = 706)**	**49(7)**	**23(3)**	**634(90)**	**53(7)**	**6(1%)**	**647(92)**	**18(3)**	**688(97)**	**9(1)**	**697(99)**	**9(1)**	**697(99)**

**Abbreviations**: **MSI**—Microsatellite instability; **IHC**—Immunohistochemistry; **Indeter**—Indeterminate; **Dmmr**—Deficient mismatch repair system; **MSS**—Microsatellite-stable.

**Table 2 cancers-14-04550-t002:** Stratification of microsatellite instability results by immunohistochemistry (IHC).

Immunohistochemistry (IHC)		Microsatellite Instability (MSI)			
	Loss of MMR	MSI-High (%)	MSI-Low (%)	MSS (%)	Total (%)
**dMMR**		**44(90) ***	**3(13) ****	**6(1) *****	**53(8)**
	MLH1/PMS2	26(53) *	2(9) **	1(0) ***	29(4)
	MSH2/MSH6	10(20) *	0(0)	3(0) ***	13(2)
	MLH1/MSH2	1(2) *	0(0)	0(0)	1(0)
	MSH2	3(6) *	0(0)	0(0)	3(0)
	MSH6	1(2) *	1(4) **	2(0) ***	4(1)
	PMS2	3(6) *	0(0)	0(0)	3(0)
**Indeterminate**		**2(4) ****	**1(4) ****	**3(0) ****	**6(1)s**
**Proficient**		**3(6) *****	**19(83)**	**625(99) ***	**647(92)**
**Total**		**49(100)**	**23(100)**	**634(100)**	**706(100)**

* Concordant, ** Minor Discordance, *** Major Discordance. **Legend**. **Abbreviations: MSI**—Microsatellite instability; **IHC**—Immunohistochemistry; **dMMR**—Deficient mismatch repair system; **MSS**—Microsatellite-stable.

**Table 3 cancers-14-04550-t003:** Cases with major and minor discordance.

Cases with Major Discordance			
MSI Analysis	IHC Analysis	NGS Results (Interpretation)	Tumor Histology
**MSI-High**	pMMR	NM_000179.2(MSH6): c.818G > T p.G273V (germline VUS)	Endometrial Carcinoma
**MSI-High**	pMMR	***MLH1* Promoter Methylation (somatic epigenetic silencing)**	Endometrial Carcinoma
**MSI-High**	pMMR	**NM_000179.2(MSH6): c.1483C > T p.R495Ter (somatic pathogenic null)** **NM_000179.2(MSH6): c.3577_3581del. p.E1193fs*2 (germline pathogenic null)**	Urothelial Carcinoma
MSS	**dMMR (Loss of MSH6)**	No mutations	Endometrial Carcinoma
MSS	**dMMR (Loss of MSH6)**	**NM_000179.2(MSH6): c.3037_3038dupAA.p.K1014fs (somatic pathogenic null)**	Neuroendocrine Carcinoma
MSS	**dMMR (Loss of MSH2/MSH6)**	NM_000179.2(MSH6): c.3260C > G p.P1087R (germline VUS)	Pancreatic Carcinoma
MSS	**dMMR (Loss of MSH2/MSH6)**	No mutations	Prostate Adenocarcinoma
MSS	**dMMR (Loss of MLH1/PMS2)**	No mutations	Epithelioid neoplasm of unknown primary
MSS	**dMMR (Loss of MSH2/MSH6**	No mutations	Adenocarcinoma of unknown primary
**CASES WITH MINOR DISCORDANCE**			
**MSI-High**	Indeterminate (questionable weak expression of MSH2 and MSH6)	**NM_000179.2(MSH6)c.2859del p.E953fs (somatic pathogenic null)** NM_000179.2(MSH6):c.1444C > pR482Ter (somatic pathogenic null) **NM_000179.2(MSH6): c.3984_3987dupGTCA p.L1330fs*12 (germline pathogenic null)**	Prostate Adenocarcinoma
**MSI-High**	Indeterminate (questionable MSH6 expression and IHC staining was inadequate)	NM_000251.2(MSH2): c.1231A > T p.I411L (somatic VUS) NM_000251.2(MSH2): c.214G > T p.A72S (somatic VUS)	Endometrial Carcinoma
MSI-Low	**dMMR (Loss of MLH1/PMS2)**	***MLH1* Promoter Methylation (somatic epigenetic silencing)**	Endometrial Carcinoma
MSI-Low	**dMMR (Loss of MSH6)**	No mutations	Colorectal Carcinoma
MSI-Low	**dMMR (Loss of MLH1/PMS2)**	***MLH1* Promoter Methylation (somatic epigenetic silencing)**	Endometrial Carcinoma
MSI-Low	Indeterminate (questionable MSH6 loss)	No mutations	Colorectal Carcinoma
MSS	Indeterminate (strong cytoplasmic positivity for MLH1, but nuclei were negative)	No mutations	Melanoma
MSS	Indeterminate (focal weak nuclear staining for MSH2/MSH6 but no positive internal control)	No mutations	Ovarian Carcinoma
MSS	Indeterminate (30% of cells were negative for MLH1 and PMS2)	NM_000251.3(MSH2):c.74G > A. p.G25D (germline VUS)	Ovarian Carcinoma

**Abbreviations: MSI**—Microsatellite instability; **IHC**—Immunohistochemistry; **dMMR**—Deficient mismatch repair system; **MSS**—Microsatellite-stable; NGS—Next-generation sequencing; **VUS**—Variation of uncertain significance.

**Table 4 cancers-14-04550-t004:** Distribution of genetic/epigenetic aberrations according to IHC pattern and MSI analysis status.

	MMR Gene Mutations (and Epigenetic Changes)												
	Somatic/Germline												
	MLH1 Methylation and Null Mutations						Missense Mutations						
	*MLH1* Methylation	*MLH1-Null*	*MSH2-Null*	*MSH6-Null*	*PMS2-Null*	Total *Null*	*MLH1-VUS*	*MSH2-VUS*	*MSH6 VUS*	*PMS2-VUS*	*MSH2-LP*	Total Missense	All Cases
IHC Patterns													
dMMR	19/**0**	2/**0**	6/**3**	3/**1**	2/**1**	30/**5**	0/**0**	1/**0**	1/**1**	2/**0**	1/**0**	5/**0**	*53*
MLH1/PMS2	19/**0**	2/**0**	0/**0**	1/**1**	0/**0**	* 20/**1**	0/**0**	0/**0**	0/**0**	1/**0**	0/**0**	1/**0**	*29*
MSH2/MSH6	0/**0**	0/**0**	5/**1**	1/**0**	0/**0**	6/**1**	0/**0**	0/**0**	0/**1**	0/**0**	1/**0**	2/**0**	*13*
MLH1/MSH2	0/**0**	0/**0**	0/**0**	0/**0**	0/**0**	0/**0**	0/**0**	1/**0**	0/**0**	0/**0**	0/**0**	0/**0**	*1*
MSH2	0/**0**	0/**0**	1/**2**	0/**0**	0/**0**	1/**2**	0/**0**	0/**0**	1/**0**	0/**0**	0/**0**	1/**0**	*3*
MSH6	0/**0**	0/**0**	0/**0**	1/**0**	0/**0**	1/**0**	0/**0**	0/**0**	0/**0**	0/**0**	0/**0**	0/**0**	*4*
PMS2	0/**0**	0/**0**	0/**0**	0/**0**	2/**1**	2/**1**	0/**0**	0/**0**	0/**0**	1/**0**	0/**0**	1/**0**	*3*
Indeterminate	0/**0**	0/**0**	0/**0**	1/**1**	0/**0**	1/**1**	0/**0**	1/**1**	0/**0**	0/**0**	0/**0**	1/**0**	*6*
Proficient	1/**0**	0/**0**	2/**0**	2/**1**	1/**0**	6/**1**	3/**8**	8/**7**	6/**9**	0/**5**	0/**0**	17/**30**	647
All	20/**0**	2/**0**	8/**3**	6/**3**	3/**1**	37/**7**	3/**8**	10/**8**	7/**10**	2/**5**	1/**0**	23/**31**	706
**MSI Analysis**													
MSI-High	18/**0**	2/**0**	6/**3**	4/**3**	2/**1**	* 30/**7**	0/**0**	2/**0**	1/**1**	2/**0**	1/**0**	6/**1**	*49*
MSI-Low	2/**0**	0/**0**	0/**0**	0/**0**	0/**0**	2/**0**	0/**1**	2/**0**	0/**0**	0/**0**	0/**0**	2/**1**	*23*
MSS	0/**0**	0/**0**	2/**0**	2/**0**	1/**0**	5/**0**	3/**7**	6/**8**	6/**9**	0/**5**	0/**0**	15/**29**	*634*
All	20/**0** *	2/**0**	8/**3**	6/	3/**1**	37/**7**	3/**8**	10/**8**	7/**10**	2/**5**	1/**0**	23/**31**	*706*

* One case had both *MLH1* methylation and *MLH1* null mutation, and one case had both *MLH1* methylation and *MSH6* null mutation.

## Data Availability

The data presented in this study are available in the Appendix A.

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
