# Peer review of "Clinical Testing for Mismatch Repair in Neoplasms Using Multiple Laboratory Methods"

_cancers, 2022, doi:10.3390/cancers14194550_

Round 1
Reviewer 1 Report
The manuscript titled “Clinical testing for mismatch repair in neoplasms using multiple laboratory methods” describes the authors reviewed 706 patients with neoplasms and further assessed by IHC staining, PCR-based MSI and NGS assays to analyze the mismatch repair conditions of neoplasms. The followings are some concerns and comments have been pointed out that the authors may want to consider.
1) Lines 24-25: Please be consistent with or without a space before and after the signs, for example, “=”, “<”, etc.
2) Line 29, line 32 and so on: please use italic p as it refers to a p-value. Please check throughout the manuscript.
3) Line 108: Please extend “MDACC”.
4) Line 113: Please include the DNA extraction method.
5) Line 114: Please include both PCR and capillary electrophoresis methods.
6) Lines 115-117: Please include brief background information why those seven markers.
7) Lines 122-123: Please specify “normal expression”.
8) Line 124: Please delete an extra full stop sign.
9) Line 125: Please include a brief protocol for methylation-specific PCR.
10) Lines 157-160: I’d suggest the authors consistent with the “case size + percentage” format descript the result which is great for understanding. Please check throughout the manuscript.
11) Discussion section line 28: A space is needed before the word “however”.
12) Please provide some IHC images and capillary electrophoresis gel images as supporting data.
13) The authors listed several questions that would be answered in the introduction section which is good. I’d highly suggest the authors mark the questions in order “1,2,…”. Then answer them in the discussion section in order to make it clearer and easier to track by readers.
Reviewer 2 Report
In general, Testing of tumor tissues for the presence of MMR gene deficiency is a standard practice in clinical oncology, with immunohistochemistry and PCR-based microsatellite instability analysis used as screening tests.
In this manuscript, Yang et al., investigated single center retrospective study on neoplasms using multiple laboratory techniques. However I have some points that muse be addressed before paper will be suitable for publication.
In the methodology, case selection should be in detailed with demographic and clinical data such as including age, gender, ethnic/sub-ethnic group, history of malignancy, tumur anatomical site and clinical cancer stage (if available) in all cases.
In the text supplimentary section was mentioned but file is not included
IHC analysis need to be explained in detail on methodology section as well as include representative figure of h&e staining and also specific markers staining of different cancer types, include antibody cat.# and concentration details.
need to be updated PCR primer information
outdated experiments were analysed and include correlation analysis of multiple experiements.
The manuscript is well written but could use some proofreading and should be accepted after major revision.
spell check on
Line 19 - variants not variangts
Line 35- staining not straining
line 213 - file is not included but in the text supplementary results and table1 mentioned ???
Round 2
Reviewer 1 Report
It’s a well-prepared manuscript. I do not have any further concerns now, except for the following minor comments that the authors should consider and double-check to homogenous the format throughout the manuscript again before publication. Good luck.
Line 156: The “10XPCR buffer”, sign is not correct. Please consider using the one I input here, “×”.
Reviewer 2 Report
Author have addressed all my comments and manuscript should be suitable for publication.